# Lumbosacral Foraminal Injections in Dogs: Preliminary Assessment of an Ultrasound- and Fluoroscopy-Guided Technique in a Cadaveric Model

**DOI:** 10.3390/ani15202958

**Published:** 2025-10-13

**Authors:** Roger Medina-Serra, Francisco Gil-Cano, Marta Soler, Francisco G. Laredo, Eliseo Belda

**Affiliations:** 1Anaesthesia and Pain Management, North Downs Specialist Referrals, Part of Linnaeus Veterinary Limited, Bletchingley RH1 4QP, UK; 2Escuela Internacional de Doctorado de la Universidad de Murcia, Programa en Ciencias Veterinarias, Universidad de Murcia, 30100 Murcia, Spain; 3Department of Anatomy and Comparative Pathological Anatomy, Veterinary Faculty, University of Murcia, 30100 Murcia, Spain; cano@um.es; 4Departamento de Medicina y Cirugía Animal, Facultad de Veterinaria, Universidad de Murcia, 30100 Murcia, Spain; mtasoler@um.es (M.S.); laredo@um.es (F.G.L.); 5Hospital Veterinario, Universidad de Murcia, 30100 Murcia, Spain

**Keywords:** foraminal injections, transforaminal epidural, lumbosacral pain, radicular pain, dorsal root ganglion, ultrasound guidance, fluoroscopy guidance, pain management, dog

## Abstract

**Simple Summary:**

Lumbosacral pain in both humans and dogs is commonly linked to affection of the lumbosacral spinal nerve. Therapeutic injections near this nerve (foraminal region) are traditionally termed “transforaminal” epidural injections because the solution travels through the foramen (a bony opening through which the nerve exits the spinal canal). This cadaveric study explored a foraminal injection technique in dog cadavers, combining ultrasound and fluoroscopy guidance to position the needle at the lumbosacral foramen and then perform an injection. We found that the technique consistently placed the injectate near the target nerve, suggesting it could help treat dogs with nerve-related lumbosacral pain. Although the injectate occasionally reached other compartments, vascular entry was rare. These results suggest this approach could represent a viable therapeutic option, pending validation in live animals.

**Abstract:**

Lumbosacral radiculopathy is a frequent cause of lumbosacral pain in both dogs and humans. Targeted lumbosacral foraminal perineural injections (commonly referred to as transforaminal epidural injections) are described in dogs and are widely used in medicine to treat lumbosacral radicular pain. This cadaveric study evaluated the injectate distribution achieved by lumbosacral foraminal injections using a combined ultrasound- and fluoroscopy-guided technique to position the tip of the needle at the cranial aspect of the foramen. Ten injections were performed in five dog cadavers using a contrast-dye mixture, and distribution was assessed by fluoroscopy, CT imaging, and anatomical dissections. Perineural epidural staining of L7 at the foraminal region was achieved in 90% of injections, with transforaminal epidural spread medial to the intervertebral foramen in 80% of injections. Subarachnoid spread occurred in 50–60%, while vascular uptake was uncommon (10–20%). The technique enabled consistent needle placement, even when nerve visualisation was limited. These findings indicate that the method can reliably achieve perineural epidural staining of L7 while minimising vascular uptake, supporting its potential clinical utility for targeted drug delivery in dogs with lumbosacral radiculopathy. Further research is needed to validate safety and efficacy in live patients.

## 1. Introduction

Lumbosacral radiculopathy is a common cause of back pain and functional impairment in both humans and dogs [1,2]. In both species, it is frequently associated with intervertebral disc herniation and foraminal stenosis [1,2], which may lead to mechanical compression, inflammation of nerve roots, and/or vascular congestion, ultimately resulting in pain [3]. In dogs, multivariable analysis has identified lumbosacral radiculopathy as the strongest predictor of lumbosacral pain [2].

Foraminal injections, commonly referred to as transforaminal epidural steroid injections, and used alongside other terms such as “selective nerve root blocks” and “periradicular injections”, are a cornerstone of interventional pain management in medicine [4,5,6,7,8,9,10]. The term “transforaminal epidural injection” is widely used in the literature to describe a range of image-guided techniques, including both early and contemporary approaches. Despite variations in technique and needle tip positioning, whether within or lateral to the foramen, studies demonstrate that foraminal injections predominantly result in epidural spread [11,12,13,14,15]. The term therefore refers primarily to the intended and observed epidural distribution of the injectate rather than to a specific needle placement or approach.

Transforaminal epidural injections are widely employed to deliver agents directly adjacent to affected spinal nerves, thereby enhancing treatment efficacy [4,5]. Nonetheless, original and newer techniques may still lead to paravertebral, subarachnoid, or intravascular spread. Therefore, fluoroscopic guidance has long been considered an essential imaging modality to confirm accurate needle placement and allow any necessary adjustments before injecting therapeutic agents via the transforaminal epidural route [16].

Beyond delivering drugs in the vicinity of the affected neural structures, foraminal techniques also facilitate the concomitant use of minimally invasive neuromodulation techniques, such as percutaneous pulsed radiofrequency (PRF), which is regarded as both safe and effective modality for alleviating radicular pain and enhancing the therapeutic effects of transforaminal epidural corticosteroid injections [17,18,19,20,21].

In medicine, transforaminal epidural injections are traditionally performed under computed tomography (CT) or fluoroscopic guidance [16,22,23]. More recently, ultrasound-guided techniques have been developed to minimise radiation exposure, avoid iatrogenic injury to soft tissue structures, and reduce the risk of inadvertent vascular puncture [24,25,26]. Although the impact of ultrasound guidance on reducing radiation exposure during spinal interventional procedures has not been quantitatively assessed in veterinary medicine, a recent study in dogs reported radiation doses at the lower end of the range described in medicine [27]. In that study, all procedures were performed using a combination of ultrasound and fluoroscopic guidance, with ultrasound employed to enhance procedural safety and reduce radiation exposure.

In veterinary medicine, lumbar transforaminal epidural injections have been previously described in both canine cadavers and live, clinically healthy dogs, with all studies using CT guidance and reporting variable success rates and complications [28,29,30].

The aim of this study was to evaluate the application of a previously described ultrasound- and fluoroscopy- guided technique for lumbosacral foraminal injections in canine cadavers [31]. The objective was to characterise the distribution of injectate achieved with this technique using complementary imaging modalities and anatomical dissections. The study also aimed to assess the technique in terms of landmark and cannula visualisation during ultrasound guidance, and the need for cannula readjustment based on fluoroscopic imaging obtained before injection. By characterising the spread, we sought to assess the technique’s potential to enable accurate perineural and transforaminal epidural drug delivery in clinical settings. We hypothesised that this technique would result in a pattern of epidural and perineural injectate spread comparable to that reported in previous veterinary studies, while reducing the incidence of intravascular spread.

## 2. Materials and Methods

All dogs used in this study either died or were euthanised for reasons unrelated to the study and were donated to the University of Murcia through the Donation Program of the Veterinary Faculty (PDCAVetMu). Ethical approval was granted by the Ethics Committee of Animal Experimentation (CEEA-OH) and the Committee of Biosecurity in Experimentation (CBE) of the University of Murcia (registry number ES300305440012). These cadavers were also used in other studies focused on different anatomical regions, with no overlap in the areas examined here. This approach adhered to the 3R principle (replacement, reduction, and refinement), ensuring that none of the studies interfered with or altered the others. Only dogs without history of trauma or anatomical alterations were used.

The study was divided into two phases. In Phase I, foraminal ultrasound- and fluoroscopy-guided lumbosacral foraminal injections were performed using a previously described technique for pulsed radiofrequency cannula positioning [31]). In Phase II, the technique’s performance and dye distribution were assessed. Five thawed adult canine cadavers were used, with the sample size determined based on availability and aligned with similar studies [29,30].

### 2.1. Phase I: Ultrasound- and Fluoroscopy-Guided Technique

The objective of this ultrasound- and fluoroscopy-guided technique was to perform lumbosacral foraminal injections using a mixture containing tissue dye (Davidson Marking; Bradley Products, Minneapolis, MN, USA) and radiological contrast (iopromide 300 mg mL^−1^, UltraVist300, Bayer, Wuppertal, Germany). All injections were performed by a single operator (RMS) with the assistance of another operator (EB), both experienced anaesthetist in locoregional techniques. Two different dye colours (blue and green) were used, with each colour applied to a different side of the lumbosacral foramina to differentiate the stain between the left and right sides of the same spinal segment during the anatomical dissections in Phase II. Both dyes were diluted with 0.9% NaCl at a ratio of 1:7.5. The final mixture, composed of equal parts of the diluted dye solution and radiological contrast, was administered at a dose of 0.1 mL/kg. The use of green or blue dye was randomised for each site (https://www.randomizer.org, accessed on 10 April 2020), ensuring that both colours were used, with each side receiving a different colour.

Cadavers were positioned in sternal recumbency with the pelvic limbs pulled forward, and the lumbosacral region was clipped. As previously described [31], a linear transducer (SL1543, MyLab Gamma, Esaote, Florence, Italy) was positioned in sagittal plane over the lumbosacral region, and then slid laterally to visualise the “crescent-moon” appearance formed by the bony surfaces of the L7 lamina and its articular processes. The transducer was subsequently titled laterally and rotated caudally to obtain a cranio-lateral to caudo-medial parasagittal view, depicting the “mouse sign”, represented by the foraminal portion of L7 (eye), the L7 transverse process (nose), and its lamina and caudal articular process (ear). Using an in-plane technique, the tip of a radiofrequency cannula (CC Straight 20-gauge, 10 cm, 5 mm tip; Boston Scientific, Valencia, CA, USA) was advanced to the cranial aspect of the lumbosacral intervertebral foramen, in close proximity to the seventh lumbar spinal nerve. Complementary fluoroscopic views were then employed to guide and confirm cannula readjustments, with each adjustment performed under continuous ultrasonographic supervision until final positioning at the cranial aspect of the foramen was achieved. This target location was selected based on the known anatomical configuration of the circumneural and dural sleeves surrounding the spinal nerve at the lumbosacral foramen [32], which influence the spread of injectate depending on the depth and angle of cannula placement (Figure 1).

The visualisation quality of relevant bony and soft tissue landmarks, the cannula (good or poor), as well as the degree of cannula readjustment (minor or major) required after the initial ultrasound-guided placement was recorded. Once in the final position, and after checking the needle hub for the presence of blood or cerebrospinal fluid, 0.1 mL/kg of the contrast-dye solution was injected. Following the injection, two primary fluoroscopy views were utilised to assess the spread of the injectate: a dorsoventral view and a dorsolateral oblique view. A lateral view was also obtained when necessary to further assess distribution. The procedure was then repeated on the contralateral site. After each injection, fluoroscopic evaluation was performed to assess epidural, subarachnoid and intravascular spread of the injectate.

### 2.2. Phase II: Imaging Study and Anatomical Dissections

The objective of this phase was to assess the spread of the injections performed in Phase I and to compare the fluoroscopic findings with those obtained from computed tomography (CT) imaging and anatomical dissections.

#### 2.2.1. CT Study

Following the injections described in Phase I, each cadaver underwent CT imaging of the lumbosacral region while maintained in sternal recumbency. Scans were performed using a dual-slice CT scanner (General Electric HighSpeed, General Electric Health-Healthcare, Madrid, Spain). The scanning protocol included a collimator pitch of 1, slice thickness of 3 mm, and a reconstruction interval of 1 mm with a 50% overlap. Scans were acquired at 120 kVp and 100 mA. Standard reconstruction algorithms for bone and soft tissues were applied. The CT images were acquired and evaluated by a radiologist (MS) blinded to the procedures performed in Phase I. The assessment focused on the presence of contrast within the epidural space, specifically medial to the intervertebral foramen (IVF), with the side of involvement (left or right) recorded when applicable. In addition, the presence of paravertebral foraminal contrast distribution at the proximal L7 region was assessed separately for each side. The evaluation also included identification of subarachnoid contrast spread. Finally, the presence or absence of intravascular contrast was recorded.

#### 2.2.2. Anatomical Study

After CT imaging, each cadaver was frozen at −20 °C for 2 days, maintaining sternal recumbency. Then, a frozen block containing the lumbosacral region was obtained using a band saw (Sierra cinta; Mainca, Spain). This block was then transferred to an −70 °C freezer for an additional day. Subsequently, cryosections were performed using high-speed band saw with a marker system, producing transverse slices approximately 0.5 cm thick at the lumbosacral level. This technique allowed for accurate identification of relevant anatomical structures and dye stain patterns without altering anatomical barriers, structures, or dye locations.

The anatomical sections were reviewed and photographed by an anatomist (FGC), who was blinded to the procedures in Phase I, and by an anaesthetist (RMS), who had performed the procedures in Phase I and had experience in anatomical dissections and cadaveric studies. The assessment focused on identifying the presence of dye within the epidural space at the level of the DRG, specifying the side (left or right) when present. Additionally, paravertebral foraminal staining, with perineural staining at the proximal portion of L7 was assessed. This assessment was performed separately for the left and right sides.

### 2.3. Statistical Analysis

Data was entered into an Excel spreadsheet (Microsoft Excel for Mac, Version 16.8; Microsoft, Washington, DC, USA) and then transferred to statistical software (R Studio, version 2024.12.0+467). Descriptive statistics were used to summarise the data. Normality of the data was assessed using the Shapiro–Wilk test. Results were expressed as mean and standard deviation for parametric data, or median and range for non-parametric data.

## 3. Results

Five canine cadavers (two mongrels and three German Shepherd dog) were included, with a mean ± standard deviation body weight of 24.1 ± 8 kg and a body condition score of and 5.8 ± 0.8. Of these, three cadavers presented degenerative lumbosacral disease and bilateral foraminal stenosis at the time of the procedures, as diagnosed by CT.

### 3.1. Phase I: Ultrasound- and Fluoroscopy-Guided Technique

The visualisation of the bony references described in Phase II as well as the visualisation of the cannula was deemed good in all the procedures (10/10). The “mouse sign” was visualised during all procedures. However, the visualisation of the eye of the mouse (referring to the foraminal portion of the seventh lumbar spinal nerve) was considered poor in 30% of the procedures (3/10). The cannulas were readjusted in 60% of the procedures (6/10) after fluoroscopy confirmation to ensure the tip was positioned in contact with the cranial aspect of the foramen. Most readjustments were minor (5/6). In two out of three procedures where visualisation of the foraminal portion of the seventh lumbar spinal nerve was considered poor, no concurrent lumbosacral degeneration affecting the foraminal region was identified.

Fluoroscopic evaluation identified epidural contrast distribution in 8/10 injections (80%), paravertebral contrast distribution in the region of the intervertebral foramen in 10/10 (100%), intravascular contrast in 1/10 injections (10%) and subarachnoid contrast in 5/10 injections (50%). Subarachnoid contrast uptake occurred in three patients, two of which presented degenerative lumbosacral disease and bilateral foraminal stenosis. One injection demonstrated aspiration of clear-like fluid prior to administration, which was followed by subarachnoid contrast spread. In another, minimal blood was aspirated, although no vascular distribution was subsequently observed.

An example of transforaminal epidural contrast spread with standard fluoroscopic views is shown in Figure 2. Additional examples of the various contrast distribution patterns observed under fluoroscopy are presented in Figure 3.

### 3.2. Phase II: Imaging Study and Anatomical Dissections

#### 3.2.1. CT Study

Computed tomography (CT) assessment confirmed epidural contrast distribution medial to the intervertebral foramen in 8 out of 10 injections (80%). Paravertebral contrast distribution in the region of the intervertebral foramen was observed in all injections (10/10; 100%).

In three cadavers, subarachnoid contrast was identified on CT imaging. However, as the CT scans were performed only after both injections had been completed, it was not possible to determine whether the subarachnoid spread originated from one injection site or from both. Consequently, in all cases where subarachnoid spread was observed, this was conservatively recorded as a positive finding for both injection sites. A similar approach was applied to one other case in which bilateral intravascular contrast was noted. Although greater contrast enhancement on one side raised suspicion that intravascular uptake may have originated from a single injection site, definitive conclusions could not be made. Therefore, intravascular spread was also considered a positive finding for both sides in that cadaver.

#### 3.2.2. Anatomical Dissections

Anatomical dissections confirmed the presence of dye within the epidural space medial to the intervertebral foramen in 8 out of 10 injections (80%). Staining within the paravertebral foraminal region was seen in 10 out of 10 injections (100%), with perineural epidural stain of the L7 spinal nerve at the foraminal region observed in 9 out of 10 injections (90%). Figure 4 displays transverse cryosections and corresponding CT images from all five cadavers, illustrating the identified dye and contrast distribution patterns. A comparison of injectate distribution findings across fluoroscopic, computed tomography, and anatomical assessments is presented in Table 1.

## 4. Discussion

This study describes the distribution of injectate resulting from lumbosacral foraminal injections using a previously described ultrasound- and fluoroscopy-guided technique for cannula placement [31]. Overall, consistent paravertebral foraminal and epidural distributions were observed, with subarachnoid contrast uptake noted in approximate half of the injections. Intravascular injectate spread was identified fluoroscopically in one injection (10%) and attributed to two injections (20%) upon conservative CT assessment.

Previous cadaveric and clinical studies have reported variable outcomes in terms of injectate spread following foraminal injections in dogs. Kneissl et al. [28] found preferential epidural spread in 66.7% (28/44) of the injections and paravertebral foraminal spread (epaxial or hypaxial) in the remaining 33.3% (14/42). They achieved spinal nerve staining in 88.1% (37/42) of injections, either during its epidural course or extraforaminally, but also noted vascular (85.7%; 36/42) and subarachnoid (61.9%; 26/42) spread. The authors noted that needle positioning did not reliably predict injectate distribution. Variability of injectate distribution was primarily attributed to anatomical variations in soft tissue structures at the foraminal region, which are not detectable by CT. They also proposed that subarachnoid migration could result from retrograde flow through periforaminal veins back into the subarachnoid space via the perispinal venous system. Liotta et al. [29] initially reported 75% (3/4) success rate in achieving transforaminal epidural injections in dog cadavers with mild spinal pathology. However, vascular spread occurred in 75% (3/4) of the injections and subarachnoid spread in 25% (1/4) of injections. In a subsequent study involving live, clinically healthy dogs, Liotta et al. [30] achieved transforaminal epidural spread in 66.7% (4/6) of injections, with vascular spread reported in 33.3% (2/6) of the injections and subarachnoid spread in 16.6% (1/6) of the injections. In their clinical study, 60% of dogs (3/5) had mild pathology: one had mild lumbosacral disc protrusion, one mild lumbosacral spondylosis, and one a mild reduction in intervertebral space.

Our findings align with these earlier reports. In the present study, 80% of the injections demonstrated transforaminal epidural spread medial to the intervertebral foramen, and 90% perineural epidural foraminal stain of the L7 spinal nerve. The incidence of vascular spread was lower (10–20%) compared with previous studies. Subarachnoid spread was observed in 50–60% of injections, a rate higher than that reported by Liotta et al. [29,30], but lower than that described by Kneissl et al. [28]. Although the sample size precludes firm conclusions, between 66.7% (4/6; CT based) and 80% (4/5; fluoroscopy based) of subarachnoid spreads occurred in cadavers with moderate-to-severe foraminal pathology at the injection site, suggesting that local pathology may increase the risk of subarachnoid spread. Such pathology may deviate or alter normal anatomical landmarks, potentially complicating ultrasonographic and fluoroscopic interpretation and affecting needle placement and injectate distribution. By including dogs with moderate to severe lumbosacral pathology, our cohort may have contributed to a higher rate of subarachnoid spread than reported in previous studies. However, this cohort may be more clinically representative, as transforaminal epidural injections are often performed on dogs with foraminal disease.

The anatomical configuration of the lumbosacral intervertebral foramen offers a basis for understanding injectate distribution in this region (Figure 1). The proximal portion of the spinal nerve, including the nerve roots and DRG, is enclosed by a circumneural sleeve composed of two layers: an outer superficial layer formed by the thoracolumbar fascia, and an inner layer derived from the peridural membrane lining the vertebral canal [32]. This sleeve has close attachments to the facet joint capsule dorsally, the caudal vertebral notch cranioventrally, and a fibrous septum caudally, which divides the foramen into cranial and caudal sub-compartments. The spinal nerve and a spinal branch of the lumbar artery traverse the cranial sub-compartment, whilst the intervertebral vein passes through the caudal sub-compartment.

These anatomical features imply that even minor adjustments in the needle tip’s position relative to these layers can significantly affect the pathway of injectate spread. When the needle tip is positioned external to the thoracolumbar fascia, the injectate is likely to remain confined to the paravertebral space without direct contact with the spinal nerve (Figure 1b). Advancing the needle through the thoracolumbar fascia, but without piercing the peridural membrane, will keep the injectate within the paravertebral space while bringing it into direct contact with the spinal nerve (Figure 1c). Penetration or tearing of the peridural membrane, without breaching the dural sleeve, will favour transforaminal epidural spread (Figure 1d). If both the circumneural and dural sleeves are breached, the injectate is likely enter the subarachnoid space (Figure 1e). Furthermore, tearing of vascular structures at either cranial or caudal foraminal sub-compartment may result in inadvertent vascular uptake.

Injectate distribution follows the path of least resistance and may be affected by multiple factors, including the pressure generated at the cannula tip during injection, the compliance of the targeted compartment, and any iatrogenic disruption of anatomical barriers during needle advancement. Additionally, subtle needle movements or displacement of adjacent tissues, whether due to operator manipulation or injection pressure, may alter the spatial relationship between the cannula and the surrounding structures. These anatomical and procedural factors ultimately determine whether the injectate preferentially disperses into the paravertebral, transforaminal epidural, or subarachnoid compartments. It should also be recognised that the spread of injectate in cadavers may not fully represent that of live patients, as post-mortem changes in tissue integrity may influence distribution. Ultimately, beyond immediate pattern of spread, properties such as formulation, viscosity, lipid solubility, protein binding, dissociation constant, and systemic absorption may play a relevant role in how therapeutic agents distribute, persist, and exert their clinical effects.

From a clinical standpoint, one major advantage of combining ultrasound and fluoroscopy for lumbosacral foraminal injections is versatility. Ultrasonography allows real-time subtle needle adjustments, allowing the operator to visualise the needle in respect soft tissue structures and reduce the risk of inadvertent vascular and neural puncture [25]. Fluoroscopy can then verify precise needle placement and real-time contrast spread to optimise injectate distribution according to specific clinical objectives [25,26,33]. Additionally, this combined imaging approach aligns with the “as low as reasonably achievable” (ALARA) principle [34] by reducing fluoroscopy time and, consequently, radiation exposure for both patients and operators. A recent retrospective study in dogs that employed this approach reported radiation doses towards the lower end of the range described in humans for comparable procedures [27].

In this cadaveric model, we deliberately refrained repositioning the needle after the first injection to document the distribution pattern most likely to occur following initial needle placement. Clinically, this simulates the preliminary contrast study routinely performed before delivering therapeutic agents, which enables confirmation of needle positioning and refinement of placement if necessary. Aspiration or visualisation of blood or cerebrospinal fluid prior to injection is widely recommended to reduce the risk of inadvertent intravascular or subarachnoid injection. However, in our study, this did not predict safe placement, as contrast distribution was still observed within these compartments despite lack of visualisation of blood or cerebrospinal-like fluid on the needle hub. This may reflect post-mortem changes in tissue compliance and fluid characteristics, potentially masking intravascular or subarachnoid positioning.

The risk of inadvertent intravascular or subarachnoid spread should be carefully considered when performing epidural corticosteroid injections, particularly with transforaminal techniques where the needle trajectory lies in close proximity to vascular and neural structures [35]. Intravascular injection, particularly into arterial vessels, can result in embolization and lead to serious neurological complications such as spinal cord infarction or stroke, especially when particulate corticosteroids are used [36]. In this cadaveric model, the absence of pulsatile blood flow may have influenced the reported incidence of vascular events. On one hand, the lack of pulsatile vessels may have hindered their identification during the ultrasonographic approach, potentially increasing the likelihood of inadvertent vascular puncture and thereby overestimating the incidence compared with live dogs. On the other hand, post-mortem clotting within punctured vessels may have “blocked” contrast entry into the vessels, diverting it externally and underestimating intravascular spread. These opposing factors should be considered when interpreting the findings, and studies in live dogs are needed to establish the true incidence in clinical practice.

Subarachnoid injection may cause chemical meningitis, arachnoiditis, or direct neurotoxicity [36,37]. Although rare, arachnoiditis has been consistently associated with inadvertent intrathecal administration of corticosteroids or preservatives [36,37]. In medicine, however, subarachnoid injections are intentionally performed for both acute indications, such as labour analgesia, and for chronic cancer and non-cancer pain [37]. Agents commonly used for this purpose are local anaesthetics, opioids, and corticosteroids, with doses and formulations specifically adjusted to minimise neural irritation and reduce adverse events [37]. Nevertheless, the subarachnoid route carries risks, particularly when injections occur inadvertently, since epidural and intrathecal dosing differ substantially [37]. Small, dilute volumes of local anaesthetics may cause only transient sensory block, but larger doses can produce extensive spinal blockade with severe cardiovascular and respiratory compromise Similarly, excessive intrathecal opioid doses may induce respiratory depression, while inadvertent corticosteroid injection has been linked to adhesive arachnoiditis [37]. Experimental canine studies have shown that intrathecal administration of methylprednisolone, whether as a single [38] or repeated [39] injection, produced histological changes such as meningeal thickening and inflammation, despite the absence of adverse clinical signs apart from a transient motor block in dogs receiving concomitant lidocaine. By contrast, inadvertent subarachnoid injection of non-ionic water-soluble radiographic contrast media is generally regarded as safe. In addition to its use during preliminary contrast studies for transforaminal epidural injections, it is routinely employed in myelographic procedures, even at relatively high volumes, without evidence of significant neurotoxicity [38]. To reduce these risks, all epidural corticosteroid injections should be preceded by careful aspiration, performed under image guidance, and include a contrast study to confirm appropriate needle placement [35].

Beyond injectable therapies, this technique offers the potential to deliver pulsed radiofrequency targeting the dorsal root ganglion (DRG) of the seventh lumbar spinal nerve as a complementary treatment [31]. In medicine, pulsed radiofrequency has been widely employed both as a standalone analgesic modality and to enhance the therapeutic effects of transforaminal epidural injections [17,18,19,20,21]. The ability to combine precise drug delivery and neuromodulation in a single, image-guided procedure highlights the potential clinical relevance of this approach for dogs with lumbosacral radiculopathies.

Our findings on the technical aspects of this ultrasound- and fluoroscopy-guided technique are consistent with the original description [31]. In both studies, the visualisation of bony landmarks and the cannula was consistently rated as good, with reliable identification of the key anatomical references comprising the “mouse sign.” However, as previously noted, the foraminal portion of the seventh lumbar spinal nerve (“eye” of the mouse sign) was poorly visualised in a notable proportion of cases. In the present study, this occurred in 30% of procedures, closely aligning with the 33% previously described. Fluoroscopic confirmation was required to optimise cannula positioning in 60% of cases, again consistent with the 66% reported in the original study. Most adjustments were minor in both investigations, involving subtle repositioning to ensure accurate placement at the cranial aspect of the intervertebral foramen. These reinforce the fact that, although ultrasound provides valuable guidance for the initial cannula trajectory, it is not sufficiently reliable to confirm precise cannula placement. This aligns with human reports, where ultrasound is considered a valuable tool for needle guidance during lumbosacral transforaminal injections. However, fluoroscopic confirmation remains essential to ensure accurate needle positioning, regardless of the degree of difficulty encountered during the initial ultrasonographic guidance [25,26,33].

While fluoroscopic imaging remains essential for confirming final needle placement and assessing injectate spread, Doppler ultrasound has been proposed as a complementary tool for visualising vascular structures during transforaminal procedures. In the cervical spine in humans, periforaminal vessels were consistently identified using Doppler ultrasound [40]. However, in the lumbar spine, their identification proved more challenging due to acoustic shadowing from surrounding bony structures [25]. In dogs, the anatomical orientation of the iliac wings further limits the ultrasonographic window available for visualising the lumbosacral foramen. This limitation may partly explain why the foraminal portion of the L7 spinal nerve could not be consistently visualised in our study. Additionally, Doppler ultrasound could not be employed in our cadaveric model due to the absence of pulsatile blood flow, limiting its applicability in non-living tissue

An integrated approach combining fluoroscopic evaluation, computed tomography (CT) imaging, and anatomical dissections allowed for a comprehensive assessment of injectate distribution following lumbosacral foraminal injections in canine cadavers. Anatomical dissections provided clear visualisation of neural structures at the lumbosacral region. The light colouration of the epidural fat provided excellent contrast, making it straightforward to detect dye within the epidural space, particularly around the DRG. However, cryosections were not deemed sufficiently accurate for identifying subarachnoid or intravascular contrast, primarily because both compartments are fluid-filled, which could lead to dye dilution and reduced visibility. In addition, the subarachnoid space’s small volume and fine dimensions could further complicate dye detection, while the presence of blood (whether fluid or clotted) could obscure dye in the intravascular compartment. Consequently, no assessment of injectate spread into these compartments was performed on the cryosections in this study.

In contrast, CT provided a clear view of contrast-enhanced injectate within the epidural space, the paravertebral foraminal region, as well as the subarachnoid and intravascular compartments. Despite this advantage, CT could not reliably determine whether the contrast had predominantly originated from one injection site or the contralateral one. Although differences in the degree of contrast enhancement raised suspicion that contrast distribution may have predominantly originated from one side, definitive conclusions regarding the laterality of injectate spread could not be drawn. As a result, in cases where subarachnoid or intravascular spread was identified in both injection sites on CT, this was conservatively recorded as a positive finding for both injection sites.

Anatomical dissections were instrumental in overcoming this limitation. The use of different dye colours for each injection site allowed clear side-specific identification of injectate distribution at each contralateral foramen. This approach enabled precise mapping of the epidural and perineural spread associated with each individual injection, which was not possible through CT imaging alone. The preservation of anatomical relationships through cryosections further minimised the risk of artefactual dye spread, enhancing the accuracy of these assessments.

Fluoroscopy proved to be a reliable tool for the identification of injectate spread during lumbosacral foraminal injections, consistently enabling detection of epidural, intravascular, and subarachnoid contrast distribution. Discrepancies between fluoroscopy and computed tomography were limited to a single mismatch for intravascular spread and a single mismatch for subarachnoid spread. These differences most likely reflect the conservative approach adopted in computed tomography interpretation, which coded laterality as bilateral when uncertain and probably overattributed intravascular and subarachnoid events.

Each imaging method offered distinct strengths and revealed specific limitations, underscoring the importance of a multimodal approach for accurately assessing lumbosacral foraminal injections. Overall, the complementary use of intra-procedural fluoroscopy, post-procedural CT imaging, and anatomical dissections provided a detailed and accurate characterisation of injectate distribution in this cadaveric model. The main limitations of this work are the small sample size and the cadaveric nature of the model, and further evaluation in live patients is required to determine the true incidence and clinical significance of subarachnoid and intravascular events, as well as to confirm the safety and efficacy of this technique.

## 5. Conclusions

This cadaveric study demonstrated that the combined ultrasound- and fluoroscopy-guided technique reliably achieved perineural L7 foraminal and transforaminal epidural injectate spread. Given that L7 radiculopathy is considered the strongest predictor of lumbosacral pain in dogs, our findings support the clinical relevance of this approach for targeted therapy. Intravascular uptake was infrequent, indicating a low risk of vascular injection. Although subarachnoid spread was higher than anticipated, the results reflect the distribution likely to occur upon initial needle placement. In a clinical setting, our results would correspond to the preliminary contrast study, which allows for fine-tuning of cannula positioning to ensure the desired spread before therapeutic injection.

## Figures and Tables

**Figure 1 animals-15-02958-f001:**
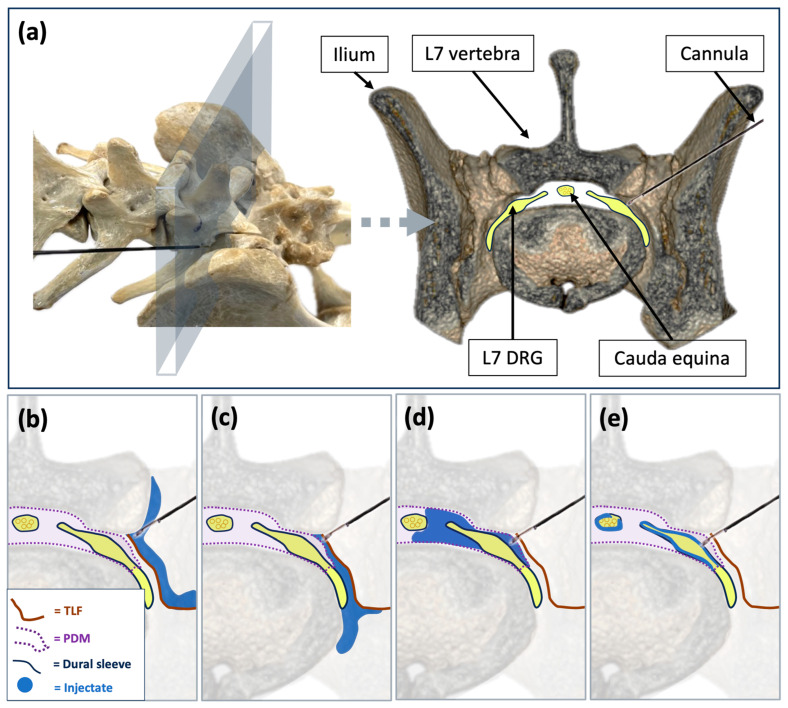
Anatomical basis and theoretical spread patterns of injectate following lumbosacral foraminal injections in dogs. (**a**) Dorsolateral oblique view of the canine lumbosacral spine and the equivalent transverse section with a cannula positioned at the cranial aspect of the left L7 intervertebral foramen. Theoretical illustrations of injectate spread based on needle position: (**b**) paravertebral distribution external to the circumneural sleeve; (**c**) paravertebral distribution in direct contact with the spinal nerve; (**d**) transforaminal epidural spread; (**e**) subarachnoid spread. DRG, dorsal root ganglion; L7, seventh spinal nerve; PDM, peridural membrane; TLF, thoracolumbar fascia.

**Figure 2 animals-15-02958-f002:**
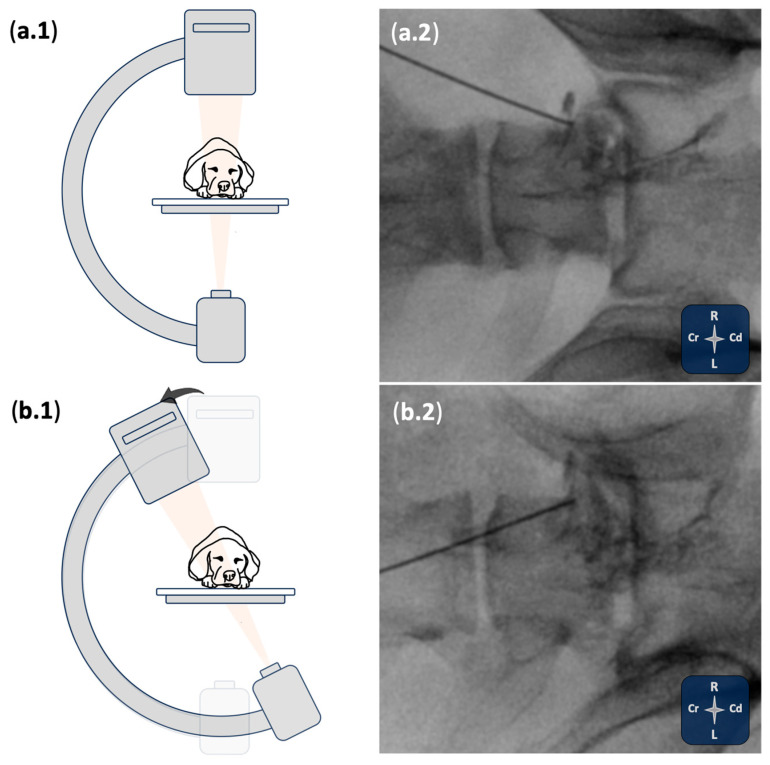
Schematic illustration of C-arm positioning and corresponding real fluoroscopic images in cadaver 2 following a foraminal image-guided injection. (**a.1**,**b.1**) Ventrodorsal (**a.1**) and ventrolateral oblique (**b.1**) C-arm positioning. (**a.2**) Ventrodorsal and (**b.2**) ventrolateral oblique fluoroscopic images showing contrast distribution consistent with a transforaminal epidural injection at the lumbosacral region.

**Figure 3 animals-15-02958-f003:**
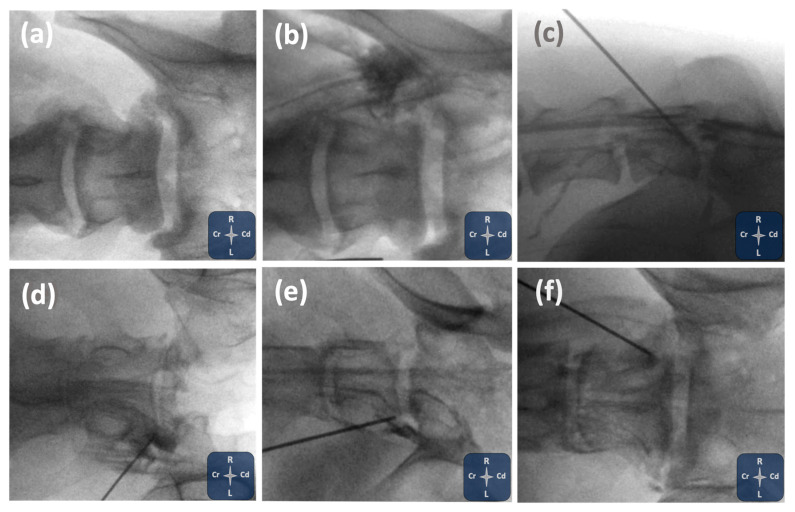
Fluoroscopic images illustrating different contrast distribution patterns in this study. Ventrodorsal views (**a**) pre- and (**b**) post-injection in cadaver 1 showing paravertebral spread of the injectate at the injection site. (**c**) Lateral view in Cadaver 4 demonstrating intravascular contrast uptake. (**d**–**f**) Ventrodorsal views in cadavers 3, 4, and 5 showing combined paravertebral, epidural, and subarachnoid contrast distribution, with predominant epidural uptake in cadavers (**d**) 3 and (**f**) 5 and (**e**) subarachnoid in cadaver 4.

**Figure 4 animals-15-02958-f004:**
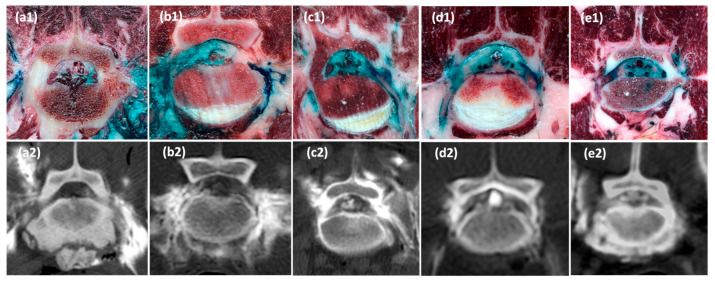
Transverse cryosections (**a1**–**e1**) and corresponding transverse CT images (**a2**–**e2**) at the lumbosacral region from the five cadavers following ultrasound and fluoroscopy guided lumbosacral foraminal injections. Epidural spread of the injectate medial to the intervertebral foramen was present in all cases except the right injection in cadaver 1 (**a1**,**a2**) and the left injection in cadaver 2 (**b1**,**b2**). Perineural stain of L7 at the foraminal region was present in all injections except the right injection in cadaver 1 (**a1**). Paravertebral foraminal contrast was present in all injections. Subarachnoid contrast is visible in cadavers 3 (**c2**), 4 (**d2**), and 5 (**e2**).

**Table 1 animals-15-02958-t001:** Comparison of injectate distribution findings across fluoroscopic, computed tomography (CT), and anatomical dissection assessments. Data represent the number of injection sites with observed distribution patterns out of the total number of injections performed (*n* = 10), with corresponding percentages in parentheses. * subarachnoid contrast was identified in three cadavers on CT imaging. As CT scans were performed after both injections had been completed, laterality could not be determined. Therefore, subarachnoid spread was conservatively recorded as a positive finding for both injection sites in these cadavers. ** bilateral intravascular contrast was identified in one cadaver. Asymmetry in contrast enhancement suggested intravascular spread originating from one injection site, although this could not be confirmed solely by CT.

Assessment Method	Epidural Contrast (n = 10)	Perineural Epidural Stain of L7 at Foraminal Region (n = 10)	Paravertebral Foraminal Contrast (n = 10)	Subarachnoid Contrast (n = 10)	Intravascular Contrast (n = 10)
Fluoroscopy	8/10 (80%)	N/A	10/10 (100%)	5/10 (50%)	1/10 (10%)
CT	8/10 (80%)	N/A	10/10 (100%)	6/10 (60%) *	2/10 (20%) **
Cryosections	8/10 (80%)	9/10 (90%)	10/10 (100%)	N/A	N/A

## Data Availability

Data supporting the findings of this study are available from the corresponding author upon reasonable request.

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
