# Peer review of "Lumbosacral Foraminal Injections in Dogs: Preliminary Assessment of an Ultrasound- and Fluoroscopy-Guided Technique in a Cadaveric Model"

_animals, 2025, doi:10.3390/ani15202958_

Round 1

Reviewer 1 Report

Comments and Suggestions for Authors

Dear Authors,

Thank you for submitting this interesting manuscript describing a combined ultrasound- and fluoroscopy-guided technique for lumbosacral foraminal injections in canine cadavers.
The study is clearly written, well-structured, and provides a comprehensive multimodal assessment (fluoroscopy, CT, and anatomical dissections). The inclusion of multiple imaging modalities strengthens the validity of the observations, and the clinical relevance is high given the rising importance of interventional pain techniques in dogs.  That said, I believe the paper would benefit from some clarifications. The small sample size is understandable but could be more explicitly acknowledged as a limitation. The relatively high incidence of subarachnoid spread is a particularly important finding and might deserve a more detailed discussion, especially regarding the possible clinical implications in live patients.
Overall, this is a valuable and well-conducted study and I congratulate the authors for it. With these small modifications, it will make a solid contribution to the field.

Author Response

Reviewer 1

We sincerely thank the reviewer for the positive feedback and constructive suggestions, which have been very valuable in improving the manuscript. We have addressed all comments in detail, and our responses are provided below.

Dear Authors,

Thank you for submitting this interesting manuscript describing a combined ultrasound- and fluoroscopy-guided technique for lumbosacral foraminal injections in canine cadavers. The study is clearly written, well-structured, and provides a comprehensive multimodal assessment (fluoroscopy, CT, and anatomical dissections). The inclusion of multiple imaging modalities strengthens the validity of the observations, and the clinical relevance is high given the rising importance of interventional pain techniques in dogs.  

Many thanks for these considerations.

That said, I believe the paper would benefit from some clarifications. The small sample size is understandable but could be more explicitly acknowledged as a limitation.

Thank you so much for this observation. It has now been explicitly stated in the conclusion that the small sample size is one of the main limitations (in line 517-520).

The relatively high incidence of subarachnoid spread is a particularly important finding and might deserve a more detailed discussion, especially regarding the possible clinical implications in live patients.

We thank the reviewer for this valuable comment. In line with this important consideration, we have expanded the section discussing the risk of subarachnoid injection to provide a more detailed evaluation of its potential clinical implications in live patients (in line 406-427).

Overall, this is a valuable and well-conducted study and I congratulate the authors for it. With these small modifications, it will make a solid contribution to the field.

Many thanks for your thoughtful comments, which have helped us improve the clarity and robustness of the manuscript.

Reviewer 2 Report

Comments and Suggestions for Authors

The manuscript addresses a highly relevant topic within the field and is aligned with the group’s ongoing line of research. The study is well-organized, and the writing is clear and fluid, with a logical flow of information. I particularly value the application of the 3Rs principles, which is an important strength of this work.

Overall, the investigation demonstrates solid scientific quality and coherence. The images provided are pertinent, well-illustrated, and accurately captioned, significantly contributing to the clarity of the manuscript.

While the paper is generally well-prepared, I have a few minor suggestions and questions that, once addressed, will further improve the manuscript and strengthen its scientific contribution. Please see my detailed comments below:

  1. Since cadaveric tissue was used, could the detection of vascular punctures be underestimated? It may be important to emphasize this point in the Results and Discussion sections.
  2. Line 33-38: I suggest describing the technique before presenting the results. For example, move the description so that it appears before the sentence “ten injections were performed…” (currently at line 38).
  3. Lines 113-115: Please include references to studies that used a similar population, ideally citing studies from other groups beyond your own, to provide broader context.
  4. Lines 131-132: Consider adding a brief summary of the ultrasound-guided technique rather than relying solely on citation number 31. Even a concise description would enhance clarity.
  5. Line 174: What is the sensitivity and specificity of CT for detecting endovenous puncture using contrast in cadavers? Could the use of cadaveric tissue reduce the detection rate of vascular punctures? Please consider addressing this in the Results and Discussion.
  6. Line 186: Please ensure consistency when referring to the anesthesiologist — use the same abbreviation (RM or RMS) throughout the manuscript.
  7. Given that the dorsal approach to the lumbosacral space is probably the most commonly used technique in interventional pain medicine for lumbosacral pain — and considering its potentially lower risk of vascular or subarachnoid puncture — it would be valuable to include a comparison of this technique in the Discussion section.

Author Response

Reviewer 2

We thank you very much for your valuable and constructive comments, which we have read with great interest. We truly appreciate your feedback, as it has helped to enrich and strengthen the manuscript. We also thank you for the time and effort spent during the revision. Please find below the answers to your comments:

The manuscript addresses a highly relevant topic within the field and is aligned with the group’s ongoing line of research. The study is well-organized, and the writing is clear and fluid, with a logical flow of information. I particularly value the application of the 3Rs principles, which is an important strength of this work. Overall, the investigation demonstrates solid scientific quality and coherence. The images provided are pertinent, well-illustrated, and accurately captioned, significantly contributing to the clarity of the manuscript. While the paper is generally well-prepared, I have a few minor suggestions and questions that, once addressed, will further improve the manuscript and strengthen its scientific contribution.

Please see my detailed comments below:

Since cadaveric tissue was used, could the detection of vascular punctures be underestimated? It may be important to emphasize this point in the Results and Discussion sections.

We thank the reviewer for raising this important observation. We believe that the use of cadaveric tissue may bias detection of vascular puncture in either direction, and we have added this section on the revised manuscript (line 396-404):

“In this cadaveric model, the absence of pulsatile blood flow may have influenced the reported incidence of vascular events. On one hand, the lack of pulsatile vessels may have hindered their identification during the ultrasonographic approach, potentially in-creasing the likelihood of inadvertent vascular puncture and thereby overestimating the incidence compared with live dogs. On the other hand, post-mortem clotting within punctured vessels may have “blocked” contrast entry into the vessels, diverting it externally and underestimating intravascular spread. These opposing factors should be considered when interpreting the findings, and studies in live dogs are needed to establish the true incidence in clinical practice.”

Line 33-38: I suggest describing the technique before presenting the results. For example, move the description so that it appears before the sentence “ten injections were performed…” (currently at line 38).

Thanks for this comment. Amended.

Lines 113-115: Please include references to studies that used a similar population, ideally citing studies from other groups beyond your own, to provide broader context.

Thank you for this comment. We have now added two references from Liotta et al., which describe a small number of transforaminal injections performed in subsets of dogs within their studies.

Lines 131-132: Consider adding a brief summary of the ultrasound-guided technique rather than relying solely on citation number 31. Even a concise description would enhance clarity.

Thanks for this comment. Following your suggestion, and to enhance clarity for the readers, we have now added a summary of the technique (in line 132-145).

Line 174: What is the sensitivity and specificity of CT for detecting endovenous puncture using contrast in cadavers? Could the use of cadaveric tissue reduce the detection rate of vascular punctures? Please consider addressing this in the Results and Discussion.

We thank the reviewer for this comment. In the clinical setting, CT is considered the gold standard for detecting intravascular contrast uptake, for example in oncology medicine or surgery, and as such represents the reference against which other imaging techniques may be compared. For this reason, sensitivity and specificity values are not usually reported for CT, and we believe this well-established principle does not require further elaboration in the manuscript. In cadaveric models, the absence of blood flow should theoretically facilitate detection by reducing dilutional effects and thereby reducing attenuation of contrast within vessels. However, this potential advantage must be balanced against the presence of post-mortem clotting, which may impede intravascular entry of contrast and give the impression of extravascular spread. This aspect has been acknowledged and discussed in the revised manuscript in lie with the previous comment from the reviewer.

Line 186: Please ensure consistency when referring to the anesthesiologist — use the same abbreviation (RM or RMS) throughout the manuscript.

Thank you so much for spotting this. Amended.

Given that the dorsal approach to the lumbosacral space is probably the most commonly used technique in interventional pain medicine for lumbosacral pain — and considering its potentially lower risk of vascular or subarachnoid puncture — it would be valuable to include a comparison of this technique in the Discussion section.

We thank the reviewer for this valuable comment. Interlaminar epidural injections are mentioned in the manuscript in the context of human data, where transforaminal injections are regarded as advantageous in patients with lumbosacral intervertebral disc herniation and radiculopathy. As the present study was specifically designed to evaluate foraminal/transforaminal epidural injections, we considered it most appropriate to focus the discussion on this technique rather than comparing it directly with interlaminar approaches.

In addition, we believe it is important to be cautious when stating that interlaminar epidural injections have a safer profile. In dogs with degenerative lumbosacral stenosis, these approaches may even be contraindicated, as dorsal displacement of the intervertebral disc can expose neural structures and the dural sac (present at this level in many dogs) to the risk of iatrogenic neural trauma and inadvertent subarachnoid injection. Moreover, only studies performed under fluoroscopy or CT guidance can provide reliable data on the incidence of subarachnoid spread in such procedures. Unfortunately, relying on the presence or absence of blood or cerebrospinal fluid upon aspiration at the needle hub has low specificity and therefore limited diagnostic value. We consider this an important yet delicate topic that merits careful evaluation in future studies.

Thank you once again for your thorough review. We are grateful for your comments, which have helped us address important concerns and improve the overall quality of the manuscript.
